# Deep Reinforcement Learning Architecture for Continuous Power Allocation in High Throughput Satellites

Juan Jose Garau Luis [1]   Markus Guerster [1]   Inigo del Portillo [1]   Edward Crawley [1]   Bruce Cameron [1]

## Abstract

In the coming years, the satellite broadband market will experience significant increases in the service demand, especially for the mobility sector, where demand is burstier. Many of the next generation of satellites will be equipped with numerous degrees of freedom in power and bandwidth allocation capabilities, making manual resource allocation impractical and inefficient. Therefore, it is desirable to automate the operation of these highly flexible satellites. This paper presents a novel power allocation approach based on Deep Reinforcement Learning (DRL) that represents the problem as continuous state and action spaces. We make use of the Proximal Policy Optimization (PPO) algorithm to optimize the allocation policy for minimum Unmet System Demand (USD) and power consumption. The performance of the algorithm is analyzed through simulations of a multibeam satellite system, which show promising results for DRL to be used as a dynamic resource allocation algorithm.

## 1. Introduction

To better serve the increasing demand of broadband data from space, next generation satellite systems will feature advanced payloads. Unlike current satellites, which present static allocations of resources, new systems will incorporate highly flexible payloads able to operate hundreds (or even thousands) of beams simultaneously and change their parameters dynamically. This increased flexibility will render invalid traditional resource allocation approaches, since these largely lean on static allocations and the use of conservative margins. Instead, satellite operators face the challenge of automating their resource allocation strategies to exploit

this flexibility and turning it into a larger service capacity.

As pointed out in (Guerster et al., 2019), dynamic resource management systems will be key to be competitive in the new markets. One key element of these systems is an optimization algorithm that computes the optimal resource allocation at any given moment. However, developing such algorithm involves dealing with a high-dimensional, non-convex (Cocco et al., 2018), and NP-hard (Aravanis et al., 2015) problem for which many classic optimization algorithms perform poorly. Multiple authors have already proposed different alternatives to overcome this problem.

Several studies have focused on approaches based on meta-heuristics, such as Simulated Annealing (Cocco et al., 2018), Genetic Algorithms (Aravanis et al., 2015; Paris et al., 2019), or Particle Swarm Optimization (Durand & Abrão, 2017). Although these algorithms have proved to be good solutions in power and bandwidth allocation problems, authors do not assess their performance under real operational time constraints. These algorithms are based on iterative methods that have a specific convergence time, which might impose a hard constraint on their real-time use.

Other authors propose approaches focused on Deep Reinforcement Learning (DRL) architectures as an alternative. DRL has already been acknowledged as a potential solution in the case of cognitive radio networks (Abbas et al., 2015), specially in multi-agent settings. Specifically, for a centralized satellite communications scenario, DRL has proved to be an operable solution for real-time and single-channel resource allocation problems (Ferreira et al., 2018). DRL also exploits the inherent time and spatial correlations of the problem (Hu et al., 2018).

However, both DRL studies propose architectures that discretize the resources before allocating them. While satellite resources such as power are intrinsically continuous, sufficient discretization might entail a notable increase in computational cost when the dimensionality of the problem is high. In this study, we explore a DRL architecture for power allocation that focuses on continuous action and state spaces, avoiding the need for discretization.

The rest of the paper is divided as follows: Section 2 describes the problem statement and the satellite communica-

---

[1]System Architecture Lab, Massachusetts Institute of Technology, Cambridge, MA, USA. Correspondence to: Juan Jose Garau Luis <garau@mit.edu>.

*Reinforcement Learning for Real Life (RL4RealLife) Workshop in the 36th International Conference on Machine Learning*, Long Beach, California, USA, 2019. Copyright 2019 by the author(s).

tions models used in this work, Section 3 presents our DRL approach, Section 4 discusses the performance of the algorithm on a simulated satellite, and finally Section 5 outlines the conclusions of the paper.

## 2. Problem Statement

This section covers, first, the motivation behind the central problem of this study; second, a detailed problem formulation introducing each of the assumptions considered; and finally, a description of the link budget model used in the following sections of the paper.

### 2.1. Problem Motivation

The next generation of satellites will allow for unprecedented parameter flexibility: the power and bandwidth, the frequency plan, and the pointing and shape of each of the beams will be individually configurable. To start exploring the adequateness of DRL to dynamically control all of these continuous parameters subject to the constraints of a real-operation scenario, in this study we only focus on one satellite resource: optimizing the power allocation for each beam while all the other parameters remain fixed.

### 2.2. Problem Formulation

We consider a multibeam GEO satellite with $N_b$ non-steerable beams, and a total available power $P_{tot}$. Furthermore, each beam has its own maximum power constraint, represented by $P_b^{max}$. For each beam, power can be dynamically allocated to satisfy the estimated demand at every time instant. The objective is to optimally allocate these resources throughout a time interval of $T$ timesteps to minimize the overall Unmet System Demand (USD) and the total power consumption.

The USD, defined as the fraction of the demand that is not satisfied by the satellite, is a popular figure of merit to quantify the goodness of a resource allocation algorithm in satellite systems (Aravanis et al., 2015; Paris et al., 2019). Mathematically, the USD is expressed as

$$USD = \sum_{b=1}^{N_b} \max[D_b - R_b(P_b), 0], \qquad (1)$$

where $D_b$ and $R_b$ correspond to the demand and data rate achieved of beam $b$, respectively. Note that there is an explicit dependency between the data-rate achieved and the power allocated to a particular beam. In other words, given a certain power allocation ($P_b$) to beam $b$, the data rate achieved ($R_b$) can be computed using the link budget equation, a procedure described in Section 2.3.

Using $USD_t$ to denote the USD attained in timestep $t$, and

$P_{b,t}$ as the power allocated to beam $b$ at timestep $t$, our optimization problem can be formulated as the following mathematical program

$$\underset{P_{b,t}}{\text{minimize}} \quad \sum_{t=1}^{T} \left[ USD_t(P_{b,t}) + \beta \sum_{b=1}^{N_b} P_{b,t} \right] \qquad (2)$$

$$\text{subject to} \quad P_{b,t} \le P_b^{max}, \quad \forall b \in \mathcal{B}, \forall t \in \{1, ..., T\} \quad (3)$$

$$\sum_{b=1}^{N_b} P_{b,t} \le P_{tot}, \quad \forall t \in \{1, ..., T\} \qquad (4)$$

$$P_{b,t} \ge 0, \quad \forall b \in \mathcal{B}, \forall t \in \{1, ..., T\} \qquad (5)$$

where $\mathcal{B}$ is the set of beams of the satellite and $\beta$ is a scaling factor. Then, on one hand, constraints (3) and (5) represent the upper and lower bounds for the power of each beam in $\mathcal{B}$ at any given timestep, respectively. On the other hand, constraint (4) expresses the limitation given by the satellite's total available power $P_{tot}$.

### 2.3. Link Budget Model

This subsection presents the link-budget equations to compute the data-rate achieved by one beam ($R_b$), assuming that a power $P_b$ has been allocated to such beam. Our link budget model is a parametric model based on (Paris et al., 2019). We only present the relevant equations to compute $R_b$ starting from a value for $P_b$, but the interested reader can find a deeper description of the elements present in a satellite communications setting in (Maral & Bousquet, 2011).

At a receiver, the link's carrier to noise spectral density ratio, $C/N_0$, quantifies the intensity of the received signal versus the noise at the receiver. A larger ratio implies a stronger signal power compared to the noise spectral density (normalized noise level relative to 1 Hz). Given the power allocation (in dB) to beam $b$ ($P_b$), $C/N_0$ can be computed as

$$\frac{C}{N_0} = P_b - \text{OBO} + G_{T_x} + G_{R_x}$$
$$- \text{FSPL} - 10 \log_{10}(kT_{sys}), \qquad [\text{dB}] \quad (6)$$

where OBO is the power-amplifier output back-off (dB), $G_{T_x}$ and $G_{R_x}$ are the transmitting and receiving antenna gains, respectively (dB), FSPL is the free-space path loss (dB), $k$ is the Boltzmann constant, and $T_{sys}$ is the system temperature (K).

With the value for $C/N_0$ we can compute the bit energy to noise ratio, $E_b/N$, a key quantity to determine whether a power allocation is valid or not, as will be explained in

Eq. (9). As opposed to $N_0$, $N$ is the noise power but not normalized to the signal's bandwidth. The link's $E_b/N$ is computed as

$$\frac{E_b}{N} = \frac{C}{N_0} \cdot \frac{BW}{R_b} \qquad (7)$$

where $BW$ is the bandwidth allocated to that beam (Hz) and $R_b$ is the link data rate achieved by beam $b$ (bps). The link data rate is in turn computed as

$$R_b = \frac{BW}{1 + \alpha_r} \cdot \Gamma\left(\frac{E_b}{N}\right), \qquad \text{[bps]} \qquad (8)$$

where $\alpha_r$ is the roll-off factor and $\Gamma$ is the spectral efficiency of the modulation and coding scheme (MODCOD) (bps/Hz), which is a function of $E_b/N$ itself. In this study, we assume that adaptive coding and modulation (ACM) strategies are used, and therefore the MODCOD used on each link is the one that provides the maximum spectral efficiency while satisfying the following condition

$$\left.\frac{E_b}{N}\right|_{\text{th}} + \gamma_m \leq \frac{E_b}{N}, \qquad \text{[dB]} \qquad (9)$$

where $\left.\frac{E_b}{N}\right|_{\text{th}}$ is the MODCOD threshold (dB), $\frac{E_b}{N}$ is the actual link energy per bit to noise ratio (dB) computed using (7), and $\gamma_m$ is the desired link margin (dB). Equation (9) validates if the resource allocation considered is feasible (i.e., there needs to be at least one MODCOD scheme such that the inequality in Eq. (9) is satisfied).

Equation (9) also allows us to compute the inverse problem, i.e. given a certain data rate we want to achieve, we can compute the necessary amount of a specific resource. Therefore, in the power allocation problem we can compute the optimal result, as an inverse problem, using (6) - (9) given the data rate required per beam ($R_b$). This means an optimization algorithm would not be needed at all. Our goal is to assess the performance of the proposed DRL architecture and compare it to the optimal actions.

Finally, in this paper we assume that the satellite use the MODCOD schemes defined in the standards DVB-S2 and DVB-S2X, and therefore the values for $\Gamma$ and $\left.\frac{E_b}{N}\right|_{\text{th}}$ are those tabulated in the DVB-S2X standard definition (ETSI EN 302 307-2, 2015). The rest of the parameters of the model, can be found in Table 1. Some of these parameters have constant values for all beams; others do not and therefore the range for each of them is showed.

## 3. Deep Reinforcement Learning Setup

This section presents, first, the general architecture of a DRL approach to solve the power allocation problem using

*Table 1.* Link Budget Parameters.

| Parameter | Value |
|-----------|-------|
| $G_{T_x}$ | 50.2 - 50.9 dB |
| $G_{R_x}$ | 39.3 - 40.0 dB |
| FSPL | 209.0 - 210.1 dB |
| $k$ | $1.38 \cdot 10^{-23}$ J/K |
| $BW$ | 655 - 800 MHz |
| $\alpha_r$ | 0.1 |
| $\gamma_m$ | 0.5 dB |

continuous state and action spaces, and second, the use of such architecture as a framework to the allocation problem specified above.

### 3.1. DRL Architecture

A basic Reinforcement Learning architecture is composed of two essential elements: an agent and an environment (Sutton & Barto, 2018). These two elements interact by means of the agent's actions and the environment states and rewards. Given a state $s_t$ that characterizes the environment at a certain timestep $t$, the goal of the agent is to take the action $a_t$ that will maximize the discounted cumulative reward $G_t$, defined as

$$G_t = \sum_{k=t}^{T} \gamma^{k-t} r_k \qquad (10)$$

where $T$ is the length of the episode, $r_k$ is the reward obtained at timestep $k$, and $\gamma$ is the discount factor. An episode is a sequence of states $\{s_0, s_1, \ldots, s_T\}$ in which the final state $s_T$ is terminal, i.e. no further action can be taken.

Figure 1 shows the specific architecture considered for the power allocation problem. The environment comprises everything that is relevant to the problem and is uncontrollable by the agent. In this case it is composed by the satellite model and the demand per beam. The agent corresponds to the processing engine that allocates power given the environment's state. Its components are an allocation policy $\pi(a_t|s_t)$, that chooses the action $a_t$ given the environment state $s_t$, and a policy optimization algorithm that constantly improves the policy based on past experience.

Since the power and demand per beam are continuous variables, the number of different states and actions is infinite. As a consequence, working with allocation policies that store the best possible action given a state is impractical. Instead, we use a neural network (NN) to model the policy and achieve a feasible mapping between an input state and an output action.

Continuous spaces also have an impact on the policy optimization algorithm. Policy Gradient methods (Sutton et al., 2000) have shown better results when states and actions

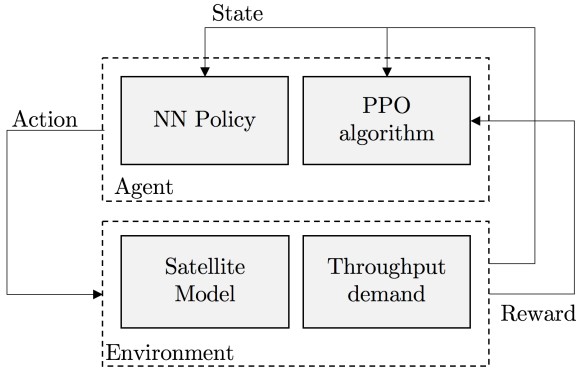

*Figure 1.* DRL architecture

are continuous spaces, as their approach focuses on directly optimizing a parametric policy $\pi_\theta(a_t|s_t)$ as opposed to computing the Q values (Sutton & Barto, 2018) and constructing a policy from them.

In this study we use a Policy Gradient method known as Proximal Policy Optimization (PPO) algorithm (Schulman et al., 2017) to improve the allocation policy. PPO algorithms derive from Trust Region Policy Optimization (TRPO) algorithms (Schulman et al., 2015) and optimize a "surrogate" objective function via stochastic gradient ascent. The algorithm tries to avoid large policy updates by clipping the objective function and using a pessimistic estimate of it. By means of this algorithm, we expect preventing changes that could make the policy perform notably worse in some cases, thus enabling more stable and less fluctuating operation of the satellite.

### 3.2. DRL Application

With the architecture presented in the previous subsection, we proceed to define its specific details. We explored different alternative state representations, one exclusively based on demand information and the other consisted of demand and past actions. We found that considering the previous optimal allocation worked best. We use $\mathcal{D}_t$ to represent the set of demand requirements per beam at timestep $t$. Then, following the approach in (Hu et al., 2018), we define the *state* of the environment at timestep $t$ as

$$s_t = \{\mathcal{D}_t, \mathcal{D}_{t-1}, \mathcal{D}_{t-2}, \mathcal{P}_{t-1}^*, \mathcal{P}_{t-2}^*\}, \qquad (11)$$

where $\mathcal{P}_{t-1}^*$ and $\mathcal{P}_{t-2}^*$ are the optimal power allocations for the two previous timesteps. Given this definition, the state is encoded using a vector with $5N_b$ components. Every time a new episode starts, the state is reset to $s_0 = \{\mathcal{D}_0, \mathbf{0}, \mathbf{0}, \mathbf{0}, \mathbf{0}\}$.

As explained in Section 2.3, since we are only using the beam powers as optimization variables, we can use (6)–(9) to determine the minimum power $P_{b,t}^*$ that satisfies $R_{b,t} \geq$

$D_{b,t}$. If for a certain beam $b$ the demand can't be met, this optimal power equals the maximum allowed power $P_b^{max}$ of such beam.

The *action* of the agent is allocating the power for each beam. Therefore, the action $a_t$ is defined as a vector with $N_b$ components, being the power values $P_{b,t}$ for each beam at timestep $t$. To respect constraints (3) and (5), these power values are clipped between zero and $P_b^{max}$. Therefore,

$$a_t = \{P_{b,t} \,|\, b \in \{1, ..., N\}, 0 \leq P_{b,t} \leq P_b^{max}\} \qquad (12)$$

As reflected in (2), the goal of the problem is to minimize the USD and the power usage during a sequence of consecutive timesteps $\{1, \ldots, T\}$. The proposed *reward* function $r_t$ focuses on both objectives and is defined as follows

$$r_t = \frac{\alpha \sum_{b=1}^{N_b} \min(R_{b,t} - D_{b,t}, 0)}{\sum_{b=1}^{N_b} D_{b,t}} - \frac{\sum_{b=1}^{N_b} (P_{b,t} - P_{b,t}^*)^2}{\sum_{b=1}^{N_b} P_{b,t}^*} \qquad (13)$$

where $\alpha$ is a weighting constant, $P_{b,t}$ is the power set by the agent, $P_{b,t}^*$ is the optimal power, $R_{b,t}$ is the data rate achieved after taking the action, and $D_{b,t}$ is the demand of beam $b$ in timestep $t$. Both the data rate and the optimal power are computed using (6)–(9).

The first element of the equation focuses on satisfying the demand while the second element responds to the necessity of reducing power without underserving that demand. Both elements are normalized by the overall demand and the total optimal power, respectively. The constant $\alpha$ is used to define a priority hierarchy between the two objectives. Given the nature of the problem, we are interested in prioritizing a smaller USD. According to the reward definition, we have $r_t \leq 0, \forall t$.

As previously introduced, Policy Gradient methods focus on optimizing parametric policies $\pi_\theta(a_t|s_t)$. In our case, the *policy* is given by the neural network, parametrized by its layers' weights. We have considered two types of networks for this study. First, we modeled the policy using a multilayer perceptron network (MLP). We found a network architecture with four layers, $15N_b$ hidden units, and ReLU activations to achieve best results in admissible training windows. We also made use of normalization layers after each hidden layer to reduce training time. The second option we studied consisted of a Long Short-Term Memory network (LSTM) with a $15N_b$-dimension array modeling the hidden state. Normalization layers were also added to the LSTM.

*Table 2.* Simulation Parameters.

| Parameter | Value |
|---|---|
| Discount factor $\gamma$ | 0.1 |
| Learning rate | 0.03 |
| Number of steps per update | 64 |
| Number of training minibatches per update | 8 |
| Number of training epochs per update | 4 |
| $\lambda$ (Schulman et al., 2017) | 0.8 |
| Clip range (Schulman et al., 2017) | 0.2 |
| Gradient norm clipping coefficient (Schulman et al., 2017) | 0.5 |
| $\alpha$ (Eq. 13) | 100 |

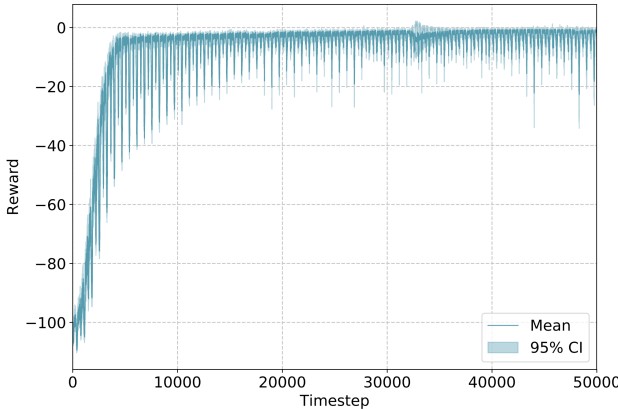

*Figure 2.* Mean and 95% confidence interval after 10 simulations of the reward sequence of the whole simulation for one environment.

## 4. Results

To assess the performance of the proposed architecture we simulate a 30-beam GEO satellite ($N_b = 30$) located over North America. For each beam, we have a time series containing 1440 data points that correspond to demand samples throughout a 48-hour activity period (a sample every 2 minutes). This data was provided by SES. Although the problem is not episodic, for computation purposes we decide to model it in a receding horizon fashion and define an episode as a complete pass through the first 720 samples of this dataset (the first 24 hours). Trying to emulate a real operation scenario, in which the agent will need to react to new data, we use the second half of the time series to evaluate the policy performance on unseen data.

Then, for each of the implemented networks, we ran 10 simulations using the parameters of the PPO algorithm listed in table 2, using batches of 64 timesteps per policy update. In all simulations we used 8 environments in parallel to acquire more experience and increase training speed. Since satisfying all customers has a higher priority than minimizing power, we observed that $\alpha$ needs to be large to obtain a desirable policy. We have used OpenAI's baselines (Dhariwal et al., 2017) for this study.

### 4.1. MLP Implementation

Figure 2 shows the mean and 95% confidence interval of the simulation reward sequence after 10 runs of 50,000 timesteps each (68 training episodes per environment, 544 in total) using the MLP policy. We can clearly observe two tendencies: First, the mean reward rapidly increases during the first thousands of iterations and then notably reduces the improvement speed for the rest of the simulation; and second, the sequence presents a high-frequency component.

Figure 3 shows the mean and 95% confidence interval, based on 10 simulations, for the aggregated *power* result of the policy during an additional episode composed by the full 48-hour dataset. The first 720 timesteps correspond to the data the policy has been trained on while the last 720 are

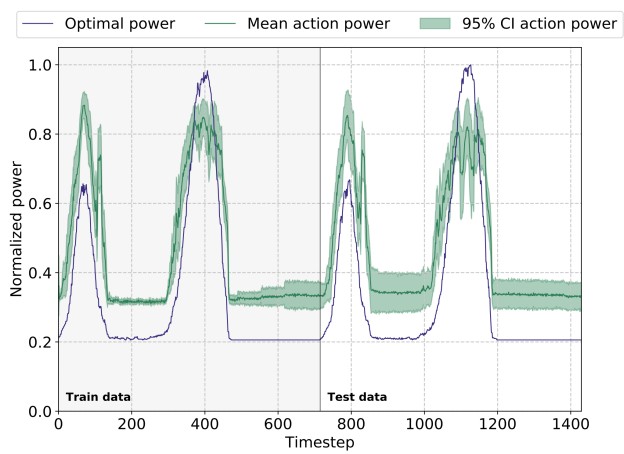

*Figure 3.* Mean and 95% confidence interval after 10 simulations of the aggregated power compared to the normalized aggregated optimal power in the MLP implementation. The last 720 samples correspond to unseen data.

unseen data. The optimal power for every timestep is also shown in the figure. The vertical axis is normalized to the maximum aggregated power value.

Figure 4 shows the aggregated *data rate* achieved using the MLP policy during the same additional episode. The aggregated demand of the dataset is also shown in the figure and the vertical axis is normalized to the maximum aggregated demand value.

We can observe the resulting policy after 50,000 timesteps responds to the demand peaks, as the data rate increases at each of them. When the demand is low, the policy sets an almost constant power and consequently sets a constant data rate at approximately 45% of the maximum demand. The variance is also larger on the unseen data.

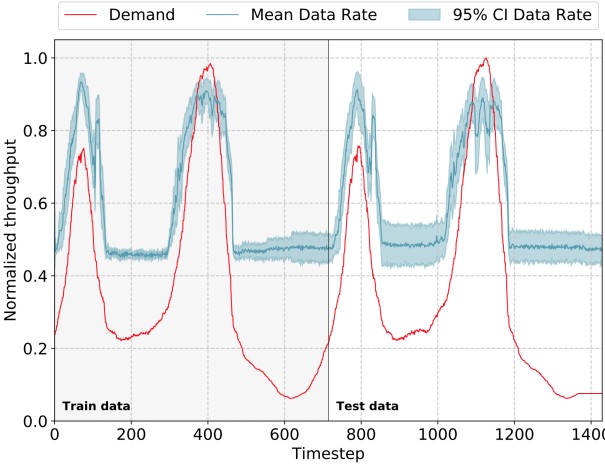

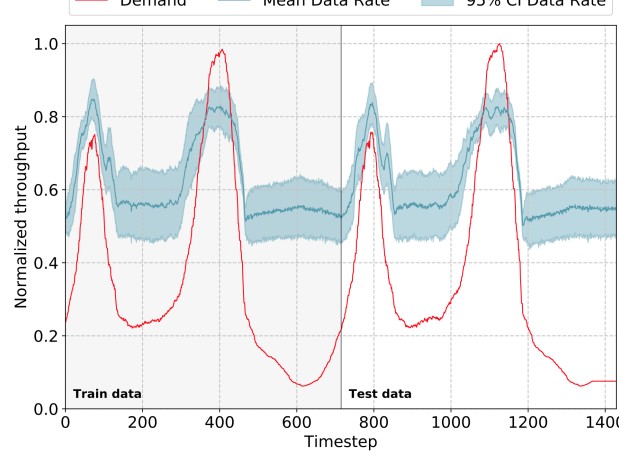

*Figure 4.* Mean and 95% confidence interval after 10 simulations of the aggregated provided data rate compared to the normalized aggregated demand in the MLP implementation. The last 720 samples correspond to unseen data.

*Figure 5.* Mean and 95% confidence interval after 10 simulations of the aggregated provided data rate compared to the normalized aggregated demand in the LSTM implementation. The last 720 samples correspond to unseen data.

Although the policy is capable of serving all demand during the first peak of the unseen data (timesteps 740 to 1000 approx.), it still shows behaviours that drift away from the desirable performance. On the one hand, although it increases power during demand peaks, it is not enough to meet the demand during the second peak and therefore the reward is penalized due to an USD greater than zero. This behaviour is repeated through all episodes and originates the high-frequency component from Figure 2.

On the other hand, during low-demand intervals, the policy achieves zero USD but is clearly allocating more power than necessary. The optimal power remains constant at a 20% while the policy sets power to 30%. The rationale behind keeping a certain power threshold, which equals to a data rate threshold, derives from the need to keep the links active as in a real scenario. In the cases where the demand is lower than the data rate threshold, the optimal power is the one that keeps the links active. If, for a certain beam, its power was to be set below this limit, such beam would become inactive and the satellite would lose capacity, since reactivating a beam requires extra capacity from the satellite.

Finally, both figures help highlighting the artifact, product of the policy, present during the second peak of the unseen data (timesteps 1050 to 1150 approx.). This type of behaviour would not be desirable during real operations, specially during demand peaks.

### 4.2. LSTM Implementation

Figure 5 shows the throughput performance of the LSTM policy. The behaviour of the LSTM policy is similar to the MLP in terms of peak response and low-demand power allocation. Comparing with Figure 4 we can appreciate the variance of the policy is larger but similar through training and unseen data. During the low-demand intervals, the data rate attained is 50-55% of the maximum demand, in contrast with the 45% achieved by the MLP policy. Finally, the LSTM policy helps to smooth the artifacts present during the second peak of the unseen data.

### 4.3. Comparison of MLP and LSTM implementations

Table 3 shows the throughput and energy performance of the MLP and LSTM policies on the unseen data, corresponding to the second day of the 48-hour dataset. Looking first at the throughput results, the demand is aggregated through all timesteps and normalized to 1. Then, the same approach is taken for the data rate, also normalized with the aggregated demand. We can observe that whilst both policies over-provide data rate, the MLP policy shows a more desirable behaviour in that sense. This preference accentuates if we compare the average USD per timestep, shown in the third row of the table.

Table 3 also shows the energy performance, defined as the power aggregation through all timesteps, on the unseen data. In this case we normalize the optimal energy to 1 and show the output energy of the policy in juxtaposition. We can see the MLP policy also shows a better result in terms of energy. When comparing both policies using the same number of hidden units ($15N_b$, 450 in these simulations), the MLP policy outperforms the LSTM; it shows both better energy and USD results. Nevertheless, the USD/demand ratio for both policies is less than 2% and therefore makes DRL a

*Table 3.* Policy performance on unseen data for MLP and LSTM implementations. Mean and 95% CI is shown.

|  | **MLP** | **LSTM** |
|---|---|---|
| Agg. demand | 1 | 1 |
| Agg. data rate | $1.68 \pm 0.15$ | $1.75 \pm 0.20$ |
| Avg. USD ($\cdot 10^{-3}$) | $9.29 \pm 5.70$ | $11.64 \pm 4.34$ |
| Max. USD | $0.20 \pm 0.10$ | $0.190 \pm 0.05$ |
| Opt. energy | 1 | 1 |
| Output energy | $1.35 \pm 0.18$ | $1.41 \pm 0.22$ |
| Avg. Eval. time (ms) | 18.6 | 20.4 |

*Table 4.* Performance of a Genetic Algorithm with different number of iterations on the power allocation problem.

|  | **Number of GA iterations** | | | |
|---|---|---|---|---|
|  | **125** | **250** | **375** | **500** |
| Agg. demand | 1 | 1 | 1 | 1 |
| Avg. USD ($\cdot 10^{-3}$) | 0 | 0 | 0 | 0 |
| Opt. energy | 1 | 1 | 1 | 1 |
| Output energy | 1.223 | 1.089 | 1.061 | 1.051 |
| Exec. time (s) | 25.6 | 49.4 | 73.9 | 98.9 |

suitable approach for the problem considered.

### 4.4. Comparison with Metaheuristics

As introduced in the beginning of this paper, the majority of previous studies on resource allocation for communication satellites lean on metaheursitic algorithms solve the optimization problem. These include Genetic Algorithms, Simulated Annealing, Particle Swarm Optimization, hybrid approaches, etc. These methods work totally opposed to DRL: while generally they do not need any previous data or training iterations, their use during real-time operations is significantly limited to their convergence time constraints.

In order to quantify the performance difference of DRL with respect to metaheuristics, we ran a simulation on the test data using a Genetic Algorithm (GA). Due to computation constraints, we took 72 samples from the unseen data, one every 20 minutes. We considered a population of 200 individuals and also used continuous variables. The results of this execution are displayed in Table 4, which shows the USD and energy performance of this method given 125, 250, 375, and 500 iterations of the algorithm. We also have included the time required to reach these results. As in the DRL case, 8 processes were used in parallel during all executions.

We can see that, although the GA achieves zero USD and better energy performance compared to any of the DRL policies, the execution time is much larger than the evaluation time of a neural network, which from Table 3 we observe is approximately 20 ms per timestep. This means running 125 iterations of the GA takes around 1,300 times more time than evaluating the DRL policies for a single timestep.

This result is directly proportional to the number of GA iterations. Given these results, a future direction to explore is the combination of DRL with one metaheuristic. Taking the almost-instantaneous evaluation of the DRL method as a starting point for a metaheuristic could produce an almost optimal performance in an admissible time window for operational purposes.

## 5. Conclusion

In this paper, a DRL-based dynamic power allocation architecture for flexible high throughput satellites has been proposed. As opposed to previous architectures (Ferreira et al., 2018; Hu et al., 2018), this approach makes use of continuous state and action spaces to compute the policy. We have set the reward function to focus on minimizing the unmet system demand (USD) and power consumption. The policy has been implemented using two approaches: an MLP network and an LSTM network.

The results obtained show, for both implementations, that the architecture produces a policy that responds to demand peaks. However, the policy is not optimal since 2% of the demand is not satisfied and an excess of energy is allocated (35% and 41% extra power using the MLP and LSTM policies, respectively). Comparing both implementations with the same number of hidden units, the MLP shows a better performance in terms of total output energy and USD. By means of a genetic algorithm analysis, we have shown that DRL is at least 1,300 times faster than metaheuristic methods, while offering comparable quality solutions (DRL performs slightly worse than metaheuristics in terms of power and USD). Based on this first study, we expect to add complexity to the problem by adding other optimization variables (bandwidth, frequency plan) into the problem. Future work will focus on the refinement and generalization of the architecture, the scalability of the policies, and the exploration of other DRL approaches.

## Acknowledgements

This work was supported by SES. The authors want to thank SES for their input to this paper and their financial support.

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
