# OpenReview forum: "Deep Reinforcement Learning Architecture for Continuous Power Allocation in High Throughput Satellites"
_ICML.cc/2019/Workshop/RL4RealLife — RL4RealLife 2019_

### Official Review · AnonReviewer2 · 2019-05-19
**Review of DRL for Continuous Power Allocation in Flexible High Throughput Satellites**

**Rating:** 3
**Confidence:** 3

**Review:**

This paper studies the dynamic power allocation problem for satellites in a setting where both states and actions are continuous. The objective is to allocate power to all beams such that the unmet demand and total power consumption are minimized at the same time. This problem is solved using Proximal Policy Optimization where the policy is given by MLP and LSTM networks (two cases).

Overall, the paper presents a new method to solve the satellite power allocation problem in a more realists setting (continuous state and action). However, the system model needs to be explained in more details for this audience. Furthermore, the reviewer is keen to see more discussion on how the result of the DRL method compares with previous work (metaheurisitcs, etc.). Since the obtained policy is not optimal, the question is whether the DRL method could still outperform other methods that do not consider continuous state and action. The data set is also rather short to explore whether the results would improve if more data were available.

Other comments:
1. Several parameters are not defined in Section 2. Is N the number of nonsteerable beams or the noise? Define N_0, C, E, and I
2. It is unclear that USD is a function of P_{b,t}. USD is a function of R_b as defined in (5) and R_b and P_{b,t} are related according to the system model. For clarity, USD should also be written as a function of P_{b,t} since it is used in (6) and throughout the paper.
3. Explain why the *two* previous demand vectors are included in the state.
4. Where did you obtain the GEO satellite data (30-beams, 48-hour data) from? Cite the paper or data source.
5. Abstract: represent —> represents

---

### Official Review · AnonReviewer1 · 2019-05-24
**Interesting application domain, but a naive baseline would be welcome.**

**Rating:** 5
**Confidence:** 4

**Review:**

Hi Authors,

I've read your paper, here are: 1) a summary in my words to make sure I understood everything and 2) comments.

This paper describes a constrained optimization problem which is then cast as an MDP.  The authors propose a both a cost function (minimize unmet system demand e.g. minimize the gap between demanded and provided throughput) as well a set of constraints (keep power under a certain threshold for each beam, as well as maintaining global power below a certain threshold).  These constraints are satisfied by clipping the actions to not allow over-powering an individual antenna.  The derived reward function encodes the constrained optimization by balancing a demand-satisfying term with a power-reduction term.  The key parameter is therefore B', which balances out these two opposing goals.  This MDP is learned using PPO.  Results are shown on a simulated satellite using both an MLP and LSTM architecture for the agent.

I find this task interesting, and in general use of RL for scheduling and control of low-power systems is an interesting domain.  Overall the paper is an easy read, and results seem promising.  There is no particular contribution to RL methodology, but I don't think that was the goal here, rather than just showing that existing RL algorithms could work in this scenario.  I was wondering why a simple baseline hardcoded controller was not used in the experiments though, it seems that a simple PID controller could potentially do a good job at controlling this task, or even applying the estimated optimal powers from timestep t-1 at timestep t.  A naive baseline would have made this paper much more compelling as the learned controller doesn't find an optimal policy from the looks of it, and so showing that it is at least better than whatever naive approach and engineer would use would really help arguing for the use of RL.

Nevertheless, I think the application domain is interesting and I would enjoy discussing more about RL & applications at the workshop so I will  support acceptance of this paper.

---

### Decision · Program_Chairs · 2019-05-28

Accept